# Vertically integrated spiking cone photoreceptor arrays for color perception

Xiangjing Wang[1], Chunsheng Chen[1], Li Zhu[2], Kailu Shi[1], Baocheng Peng[1], Yixin Zhu[1], Huiwu Mao[1], Haotian Long[1], Shuo Ke[1], Chuanyu Fu[1], Ying Zhu[1], Changjin Wan ✉[1] & Qing Wan[1,3] ✉

The cone photoreceptors in our eyes selectively transduce the natural light into spiking representations, which endows the brain with high energy-efficiency color vision. However, the cone-like device with color-selectivity and spike-encoding capability remains challenging. Here, we propose a metal oxide-based vertically integrated spiking cone photoreceptor array, which can directly transduce persistent lights into spike trains at a certain rate according to the input wavelengths. Such spiking cone photoreceptors have an ultralow power consumption of less than 400 picowatts per spike in visible light, which is very close to biological cones. In this work, lights with three wavelengths were exploited as pseudo-three-primary colors to form 'colorful' images for recognition tasks, and the device with the ability to discriminate mixed colors shows better accuracy. Our results would enable hardware spiking neural networks with biologically plausible visual perception and provide great potential for the development of dynamic vision sensors.

Color vision is an ability to perceive differences between light composed of different wavelengths, providing substantial environmental adaptivity to organisms[1]. The light passes through the cornea and lens and forms an inverted image on the retina at the back of the eye. The retina contains two types of photoreceptors: rods and cones[2]. Color information is detected in daylight by cones and transmitted to the brain for color perception[3]. Unlike the digital camera obtains color information by filters and processes based on centralized, sequential, and binary operations, a color vision formed in a biological visual system relies on cone-type photoreceptors that selectively respond to light with three wavelengths and encode them into spike trains for event-driven, temporal-correlated, and parallel processing[4–8]. As a result, the feature size and response time of the image sensors like charge-coupled device (CCD) or complementary metal oxide semiconductor (CMOS) is crucial to the efficiency of intelligent tasks such as image segmentation and object recognition[9], which inevitably requires enormous throughput as well as energy consumption[10,11]. There are three types and ~6 million cones in our eye[12,13], which

consume roughly hundreds of picowatts of each and enable us to discriminate more than 1 million colors in a very compact configuration, outperforming most of the digital sensors[14]. Hence, developing biologically plausible artificial photoreceptors, especially the cone type, would give birth to a visual system with exquisite visual perception and extremely high energy efficiency and would boom the related areas such as prosthesis[5,15,16], neurorobotics[17,18], and cyborgs[19].

Hardware spiking cone photoreceptors (SCPs) are able to respond differently to light composed of different wavelengths and encode them into spiking at a certain rate. In the beginning, the emulation of essential synaptic functions in the visual neural system and the development of light-sensitive synaptic devices were pursued, which is aimed at mimicking short-term/long-term memory with respect to light[20–22]. For example, an optic-neural synaptic device based on an optical sensing transistor and a synaptic transistor connected in series was proposed, which is able to classify color-mixed patterns with a full-connected optic-neural network[20]. More recently, the artificial spiking receptors have aroused great interest based on a consensus that

[1]School of Electronic Science and Engineering, Collaborative Innovation Center of Advanced Microstructures, Nanjing University, Nanjing 210093, China. [2]College of Integrated Circuit Science and Engineering, Nanjing University of Posts and Telecommunications, Nanjing 210003, China. [3]School of Micro Nanoelectronics, Zhejiang University, ZJU-Hangzhou Global Scientific and Technological Innovation Centre, 310027 Hangzhou, PR China. ✉ e-mail: cjwan@nju.edu.cn; wanqing@nju.edu.cn

information implied in rate is extremely energy-efficient and very robust to noise[23-30]. A spiking photoreceptor based on an optical sensor and oscillation neuron in series exhibited efficient edge image segmentation out of a complex background, representing a pioneer and feasible approach toward SCP[10]. However, a capacitor-free and more compact configuration are required for pursuing a smaller footprint. Furthermore, high biological plausibility is still challenging, and essential properties with respect to energy consumption, range of spiking rate, selectivity to different wavelengths of light, and so on have not been well realized.

Here, we demonstrate a vertically integrated SCP (VISCP), which is capable of converting light into spike trains and discriminating lights composed of different wavelengths at an ultralow energy consumption. The VISCP is built based on a vertically integrated configuration of indium−tin−oxide (ITO)/tantalum-oxide (Ta₂O₅)/Ag/indium−gallium−zinc-oxide (IGZO)/ITO. The power consumption in response to visible light is ≤400 pW, and the spiking rate of such VISCP is ~0.1–1200 Hz, which is very close to the response range of biological cones[14]. Such devices have been verified by three wavelengths of light and exhibit a high selectivity (>1.5 orders of magnitude) which enables the discriminating of different combinations of

these lights. This also facilitates the demonstration of color-blind test simulations due to such high selectivity. Handwritten digits with color-blind tests are served as the testing dataset, and the differences in recognition accuracy can be observed between devices with and without the ability to discriminate mixed colors. This oxide-based VISCP could be regarded as a building block for hardware-spiking neural networks with sophisticated color perception.

## Results

### Structure and characterization of the VISCP

In biological visual systems, photoreceptors (rod and cone cells) convert external optical stimuli into spiking potentials, which eventually form vision in the brain, as shown in Fig. 1a. Rod cells are sensitive to dim light while lacking color-distinguishing ability[31]. Cone cells, which work in bright environments, contain three types of light-sensitive pigments (red, green, and blue). Cone cells encode specific wavelengths of light into spikes with specific frequencies, which are the basis of color vision[32]. These photoreceptor cell bodies are precisely arranged in clusters in the apical region of the eye disc and project axons into the brain's optic lobe (Fig. 1b)[33]. Inspired by the cone cells, a VISCP is proposed with a device configuration of

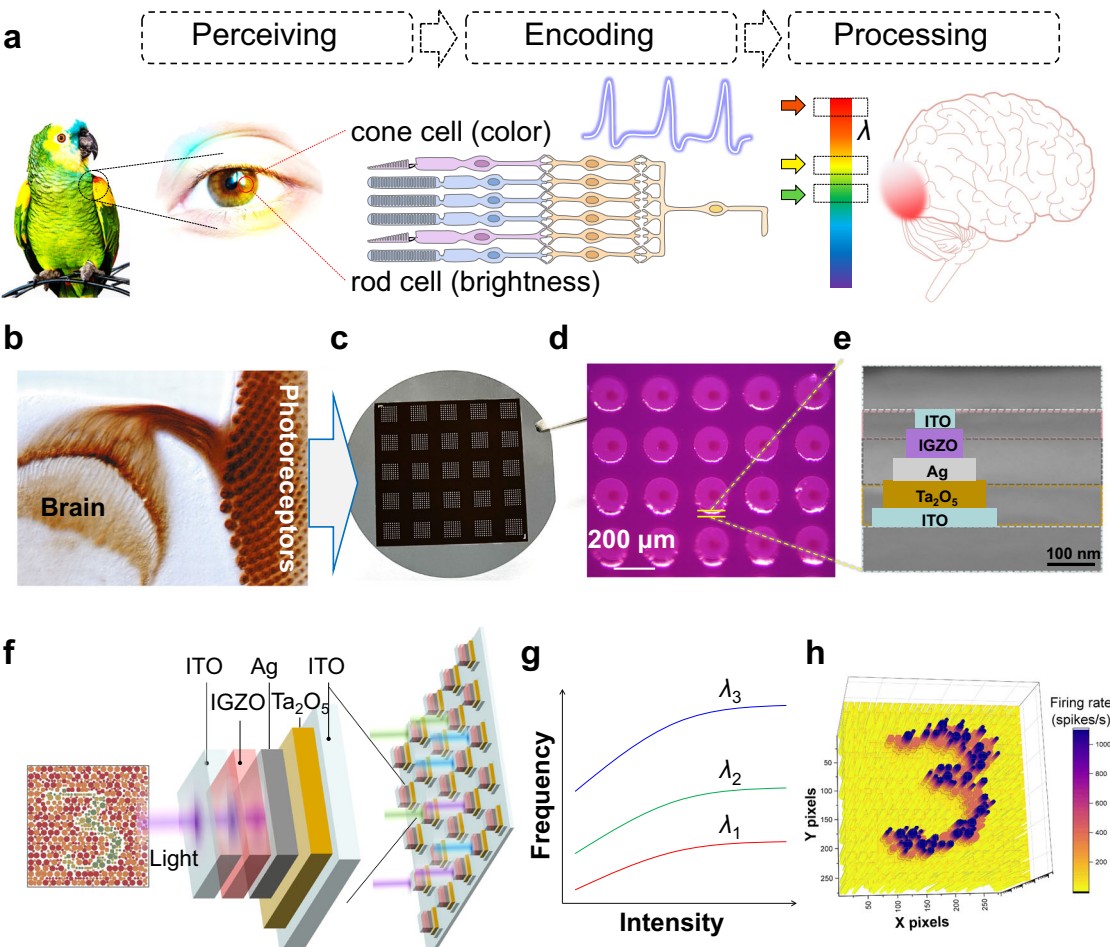

**Fig. 1 | The vertically integrated spiking cone photoreceptors (VISCP) and their biological counterparts. a** Biological photoreceptor convert specific colors of optical stimuli into spiking potentials. The coded information is ultimately sent to the brain for further processing. The Portraits of the colorful parrot and women's eye are reproduced with permission from Pexels. **b** The photoreceptor cell bodies of wild drosophila are connected to the brain's optic lobe. Reproduced with permission[33]. Copyright 2004, COMPANY OF BIOLOGISTS. **c** The digital image of oxide-based VISCP array on the two-inch silicon wafer. The whole array contains 5 × 5 sub-arrays, and each sub-array consists of 8 × 8 VISCP devices. **d** Micrograph of a group of oxide-based VISCP shows the micropillar structure. **e** The sectional view of such micropillar illustrates the vertically integrated layers. **f** Schematic illustration of color blindness image perception in artificial spiking cone photoreceptors array (ITO/ Ta₂O₅/Ag/ IGZO /ITO). **g** The spiking frequency as a function of light intensity with different wavelengths. **h** A 3D stereo image of firing rate after artificial spiking cone photoreceptors recognition.

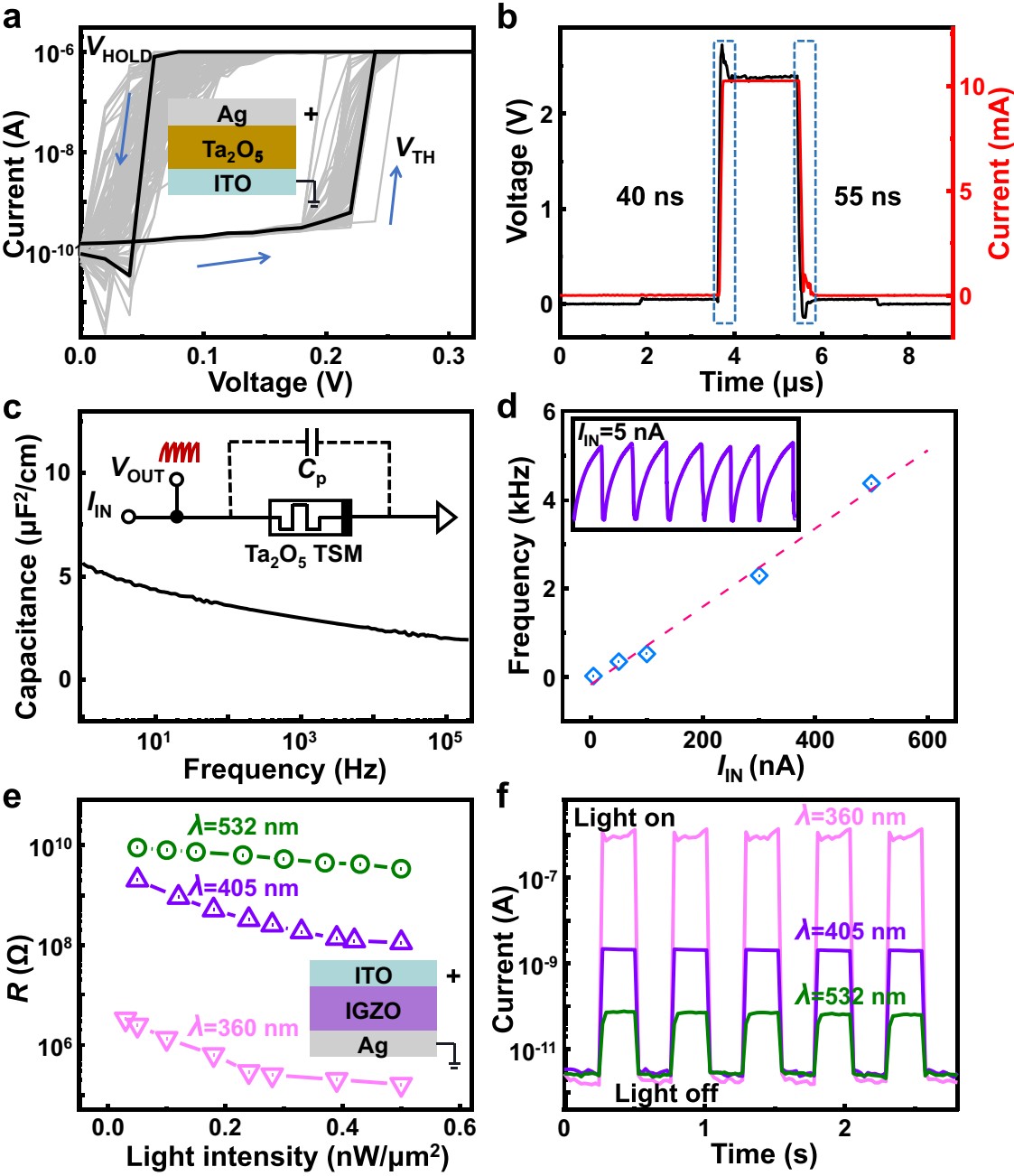

**Fig. 2 | Electrical characterizations of the photoresistor and spike-encoder.**
**a** Current–voltage curves of the ITO/Ta$_2$O$_5$/Ag memristor in 500 sweep loops.
**b** The $V$–$T$ characteristics of the memristor and the switching speed between high and low resistance states. **c** Frequency-dependent specific capacitance of the Ta$_2$O$_5$-based TS memristor with inset showing the equivalent circuit. **d** The firing frequency is a linear relationship with the $I_{IN}$, and the short-dashed line is a linear fit. **e** The resistance of the IGZO sensor as a function of light intensity with different wavelengths. Inset: schematic of Ag/IGZO/ITO sensor. **f** The transient electrical characteristics of the IGZO sensor under different wavelengths with an applied voltage of 0.2 V.

ITO/Ta$_2$O$_5$/Ag/IGZO/ITO (Fig. 1c–e), and the response to colorful lights is conceptually illustrated in Fig. 1f. The VISCPs consist of an IGZO-based photoresistor and a Ta$_2$O$_5$-based spike-encoder, which is capable of direct transducing persistent lights into spike trains dependent on the light intensity. This enables the discrimination of optic patterns with different intensities pixels similar to rod cells. While in order to mimic the properties of cone cells, a VISCP should respond differently to light with different wavelengths (e.g., the responses to wavelengths of $\lambda$1, $\lambda$2, and $\lambda$3 as shown in Fig. 1g). In this case, the colorful patterns like color-blind test image could be transformed into a pattern with high contrast ratio in terms of spiking rate, which facilitate the recognition of such pattern.

## Electrical characterizations of the spike-encoder and photoresistor

The spike-encoder is an ITO/Ta$_2$O$_5$/Ag-based threshold switching (TS) memristor, as shown in the inset of Fig. 2a. The IV characterization exhibits the typical TS property as shown in Fig. 2a. A sweep voltage was applied on the top electrode with a compliance current of 1 μA and the bottom ITO electrode was grounded. When the applied positive voltage exceeds the threshold voltage ($V_{TH}$), Ag conductive filaments (CFs) are formed to bridge the top and bottom electrodes, enabling the memristor to switch from the high resistance state (HRS) to the low resistance state (LRS). The formation of CFs is dominated by cation migration and redox processes[34–39]. When the voltage sweeps back and

is below the hold voltage ($V_{HOLD}$), CFs would rupture spontaneously due to the interfacial energy minimization[40,41], switching to the HRS of the memristor. The TS characteristics exhibit no obvious degradation during 500 consecutive cycles. The $V_{TH}$ and $V_{HOLD}$ increase with the sputtering time of the $Ta_2O_5$ layer, which is shown in the Supplementary Fig. 1. The HRS–LRS switching speed of the TS memristor is shown in Fig. 2b. The driving pulse with an amplitude of 2.5 V and duration of 2.0 μs is applied to the memristor. A fast switching-on speed of ~40 ns and a recovery time of ~55 ns after the driving pulse can be observed by applying a reading voltage with an amplitude of 0.05 V and a duration of 2.0 μs. The negative differential resistance (NDR) effect is also known as TS. The NDR effect means that as the applied current increases, the voltage decreases instead.

$$R = \frac{dV}{dI} < 0 \qquad (1)$$

In Supplementary Fig. 2, the NDR effect can be observed through current sweeping. The voltage decreases as the applied current increases, resulting in a negative resistance when the TS memristor voltage reaches the threshold voltage ($V_{TH}$). The NDR effect is attributed to the formation of the Ag filament, resulting in a sharp drop in resistance above $V_{TH}$. Such an NDR effect provides the basis for the oscillation of a spike-encoder[42,43]. The parasitic capacitance ($C_P$) of the TS memristor is estimated to be 2.5-5 μF/cm² at the frequency range between 1 and 1000 Hz, as shown in Fig. 2c. The equivalent circuit of the device is shown in the inset of Fig. 2c. The spike-encoder takes full advantage of such high parasitic capacitance, enabling a capacitor-free configuration, simplifying the structure, and offering greater potential for further scaling down.

The output spikes ($V_{OUT}$) could be observed by applying a current bias ($I_{IN}$), as shown in Fig. 2d. The current bias would charge the parasitic capacitor as long as the $Ta_2O_5$-based TS memristor is at its HRS. This charging process increases $V_{OUT}$ until it approaches the $V_{TH}$. The memristor will switch from HRS to LRS when $V_{OUT}$ increases to $V_{TH}$. Due to the reduced resistance, the parasitic capacitor discharges, and $V_{OUT}$ drops rapidly. When the voltage is below $V_{HOLD}$, the TS memristor recovers to HRS, and the $C_P$ is charged again. As a consequence, the charging/discharging process, along with the spontaneous resistance switching (HRS→LRS→HRS) of the TS memristor, underlies the oscillating in $V_{OUT}$. Supplementary Fig. 3 clearly shows the charging and discharging processes in a single spike behavior. The increase in input current would accelerate the charging process and lead to an increase in the spiking rate of $V_{OUT}$. The spiking rate is plotted as a function of the input current, as shown in Fig. 2d, exhibiting a nearly linear relationship. As the input current ($I_{IN}$) increases from 5 to 500 nA, the spiking rate of the $V_{OUT}$ increases from 25 Hz to 4 kHz. The inset shows the typical response with $I_{IN} = 5$ nA, which depicts the voltage-spiking behavior. The spiking behavior of the other input currents is shown in Supplementary Fig. 4.

The artificial photoreceptor is an Ag/IGZO/ITO-based photoresistor, as shown in Fig. 2e. The resistance state of the IGZO-based photoresistor in response to increasing light intensity under different wavelengths is also presented. The band gap of perfect IGZO films is about 3.5 eV, which can absorb high-energy photons (e.g., UV light with a wavelength of 360 nm)[44]. The introduction of oxygen vacancies in the IGZO film can be achieved by controlling the growth atmosphere (oxygen–gas partial pressure relative to argon) during the sputtering deposition process. The oxygen vacancies lead to a defect energy level lower than 3.5 eV, which enables a certain level of sensing capability to visible lights[45,46]. Supplementary Figure 5 shows the optical absorption spectra of the IGZO film. It clearly shows that the light absorption of the IGZO film decreases with increasing wavelength. Therefore, light with different wavelengths can induce significant differences in conductance. Lights with three wavelengths were used in this work. The wavelength

gaps among the three lights are large enough to enable the differentiation by the IGZO-based photoresistor, as shown in Fig. 2e. Figure 2f displays the transient response to different wavelengths with a light intensity of 0.5 nW/μm². Significant differences can be observed in the resistance state of the IGZO-based photoresistor among different wavelengths, indicating the capability of color selectivity. The IGZO photoresistors with different sputtering times are shown in Supplementary Fig. 6. It shows that the resistance increases with the thickness of the IGZO layer.

## Vertically integrated VISCP and color selectivity

The vertically integrated ITO/$Ta_2O_5$/Ag/IGZO/ITO device that incorporates the spike-encoding and light-sensing properties enables the mimicking of cone functions. The equivalent circuit of the integrated VISCP is shown in Fig. 3a. A constant voltage ($V_{Bias} = 0.5$ V) and ground voltage were applied on the top and bottom ITO electrodes, respectively. The output voltage ($V_{OUT}$) was measured on the Ag electrode. A more detailed description of the design of the VISCP is given in Supplementary Fig. 7. A detailed structural and chemical characterization of the IGZO and $Ta_2O_5$ films is presented in the Supplementary Information (Supplementary Figs. 8 and 9). The device monolithically encodes persistent light into a spike train with a certain level of frequency, as shown in Fig. 3b. The VISCPs are resting in a dark environment. In contrast, the SPCs continuously encode and fire under persistent light illumination ($\lambda = 360$ nm, $P = 0.03$ nW/μm²). The frequency of output spikes in the VISCP exhibited a positive correlation with the light intensity and wavelength (Fig. 3c). Lights with wavelengths of 360, 405, and 532 nm were used as stimulations in this work, and the oxide-based VISCP exhibits strong distinction to these 'color'. There is no overlap among the frequency ranges in response to these wavelengths. In this case, the spiking rate can convey color and intensity information of light stimulation. Figure 3d shows the experimental observation of the spiking in response to the three wavelengths with a fixed intensity of 0.5 nW/μm². The spiking frequencies of the VISCPs were 1200, 7, and 0.1 Hz for lights with wavelengths of 360, 405, and 532 nm, respectively. The effective inputs of the memristor depend on the resistance of the IGZO-based photoresistor, which is wavelength and intensity-dependent and modulates the spike frequency. For the wavelength of 360 nm, as the light intensity reduces from 0.5 to 0.03 nW/μm², the frequency decreases from 1200 Hz to 37 Hz, as shown in Fig. 3e. In conclusion, the VISCPs have color selectivity in bright environments while ineffective in the dark environments, which is similar to the biological cone.

The device-to-device variation data are shown in the Supplementary Fig. 10. The Gaussian fit of Supplementary Fig. 10a demonstrates a certain amount of variations of $V_{TH}$ (-0.27 V ± 0.06) and $V_{HOLD}$ (-0.04 V ± 0.017) (the amplitude of the output spike train). The frequency statistics of the different device responses for the three wavelengths are shown in Supplementary Fig. 10b. The spike frequencies of the VISCPs are clearly distinguished at the three illumination wavelengths. Table 1 shows a comparison of several artificial visual nerves/photoreceptors and the human eye photoreceptor[10,14,18,20,47–53]. Our work features both spike encoding and color perception. It has a low power consumption (≤400 pW per spike) in response to visible light, similar to the photoreceptors of the human eye, which well mimics the information encoding scheme of its biological counterpart.

## Color-blind image recognition

The color perception of the biological visual system depends on the cone's response to the ratio of red, green, and blue. The VISCP spike rates increase as the wavelength decreases from 532 nm ($\lambda1$) to 405 nm ($\lambda2$) and 360 nm ($\lambda3$). We utilized the three wavelengths as pseudo-colors and performed a mixed-color test as shown in Fig. 4a. Four mixed-colors defined by the power percentage of the combined lights were: 100% $\lambda1$ for 'red', 50% $\lambda1$ and 50% $\lambda2$ for 'orange', 50% $\lambda2$ and 50% $\lambda3$ for 'olive' and 100% $\lambda3$ for 'green', respectively. The total energy

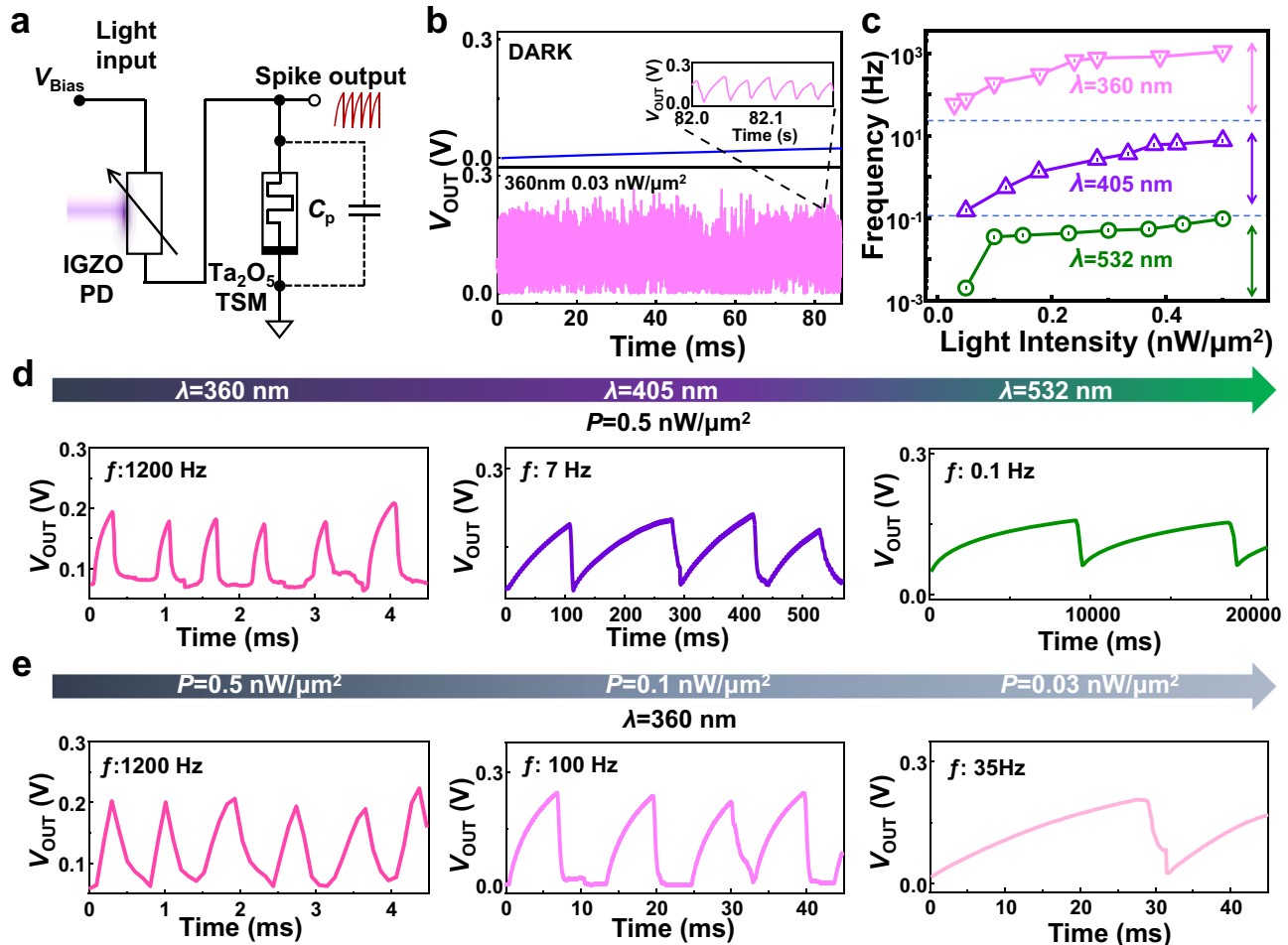

**Fig. 3 | The spike-encoding behavior and color selectivity of the vertically integrated spiking cone photoreceptors (VISCP) under light illumination. a** The equivalent circuit of the integrated VISCP. **b** The VISCP is resting in a dark condition. The VISCP fired spikes with light a wavelength of 360 nm. **c** The fire frequency plotted as a function of the light intensity with different light wavelengths. **d** Experimental observation of the VISCP was fired with three different frequencies under 360, 405, and 532 nm. The light intensity was kept constant at 0.5 nW/μm². **e** Experimental observation of the VISCP was fired under various intensities at the wavelength of 360 nm.

intensity of each light input is fixed at 0.5 nW/μm². The spike rates in response to red, orange, olive, and green lights increase exponentially, as shown in Fig. 4b. The spike rate for red light is the lowest at 0.2 Hz, while the rate reaches the highest of 1200 Hz under green light irradiation. The spike rates difference between adjacent colors is over one order of magnitude, indicating the excellent selectivity of the VISCP in distinguishing mixed colors. The modified MNIST handwritten digit images were generated on MATLAB based on the color blindness test style for image preprocessing, which consist of randomly distributed circles with several similar colors. In this work, the main body pixels of a handwritten digit were randomly painted orange and red, while the background pixels were randomly painted olive and green. In this way, a dataset for red-green color blindness, the most common type of color blindness, can be generated with a size of 280 × 280. Generally, red-green color blindness is difficult to tell the difference between red and green, especially for the mixed colors that contain red or green, like orange and olive. As shown in Fig. 4c, we simulate the behaviors of individuals with normal color vision and color blindness. The device with excellent selectivity to the four 'colors', as demonstrated in Fig. 4b, was analogous to the one with color vision. A simulated device that can only differentiate 'red' and 'green' and cannot differentiate 'red/orange' and 'green/olive' was analogous to the one with red-green color blindness. The parameters for simulation are extracted from Fig. 4b (details of simulation see Supplementary Note 3). Five thousand treated images

were mapped to the light matrixes, which can trigger the spiking responses of the array of VISCP. Then the spiking rate of each VISCP was measured, serving as the preprocessed images for further processing. The preprocessed images were fed into a five-layer convolutional neural network for recognition, with 90% of the images for training and the rest for testing (details of simulation see Supplementary Note 4). Figure 4d shows recognition accuracy during 30 training epochs for the devices with and without the mixed-color selectivity. Although the treated digits are more complex than the original MNIST digits and a relatively low recognition accuracy can be observed, the VISCP 'eye' with color selectivity can identify the target digit from the background with mixed colors with an accuracy of ~83.2%. However, the 'eye' without mixed color selectivity shows great difficulty, and a low accuracy of only 75.5% was achieved. Such results successfully mimicked the color perception of human, which incorporate sensing, rate encoding, and recognition. We also demonstrate color selectivity as a significant positive effector on the recognition accuracy of complex objects, which exists among humans with color and color-blindness vision.

## Discussion
Future apparatuses that are intended to interact with humans and/or the environment can benefit inexhaustibly from biological systems with highly sophisticated perceptual and sensorimotor capabilities. The biological counterparts enable energy-efficient and autonomous

**Table 1 | Summary of the reported basic performance of the artificial visual nerve/photoreceptors and our device**

| Artificial visual nerve/photoreceptor | Spike encoding | Spike rate | Power consumption per spike | Light response range | Vision-related functions |
|---|---|---|---|---|---|
| h-BN/Wse$_2$ photoresistor and Wse$_2$ transistor[20] | × | - | - | 405–655 nm | Colored and color-mixed pattern recognition |
| PTCDI-C8/VOPc light-sensitive element and P(VDF-TrFE)/ P(VP-EDMAEMAES) gated P(IID-BT) transistor[47] | × | - | - | 550–850 nm | light intensity and frequency transduction |
| ITO/MoO$_x$/Pd/SiO$_2$/Si[48] | × | - | - | 365 nm | Image memorization and preprocessing |
| PEA$_2$MA$_2$Pb$_3$I$_{10}$-based photoresistor & ITO transistor[18] | × | - | - | Solar light | Visual-haptic fusion |
| Ta/InGaZnO$_4$/Pt and Pt/NbOx/Ta[10] | √ | ~1.4-5 ×10$^6$ spike/s | ~1mW @365 nm ~ 2 mW @254 nm | 254–365 nm | UV image segmentation |
| Commercial photoresistor and Ag/TaO$_x$/ITO[49] | √ | 1-200 spike/s | 0.5 µW @532 nm | 532 nm | Visual depth perception |
| p-i-n perovskite optoelectronic device[50] | × | - | - | 435–700 nm | Excitatory and inhibitory light-mediated synaptic functions |
| MoS$_2$ transistor[51] | × | - | - | 660 nm | Visual adaptation |
| Ti/Au/GaO$_x$/SiO$_2$/Si and TiN/TaOx/HfO$_x$/TiN[52] | × | - | - | 254 nm | Latent fingerprint identification |
| ITO/IGZO/Ag/Ta$_2$O$_5$/ITO (This work(*)) | √ | 0.1-1200 spike/s | 210 nW @360 nm 400 pW @405 nm 4.1 pW @532 nm | 360–532 nm | Color perception |
| The photoreceptor in the human eye[14,53] | √ | ~1–1000 spike/s | 250 pW @Dark 50 pW @Bright | 400–780 nm | Visual perception |

*The power consumption estimation can be found in Supplementary information.

interactions with the real world, where the signals are always non-structural, non-normalized, and fragmented. The sensing information in conventional systems is encoded into amplitudes (analog) or represented by binary signals (digital), which are thought to be data-intensive and highly redundant. What's worse, the physical separation of sense, memory, and processing in these systems aggravates the computational burden. Encoding external stimuli into spikes could be regarded as the most biologically plausible coding scheme, and the converter that is capable of spike-encoding could be regarded as the core of a future bionic system.

In this work, the vertical-integrated oxide-based devices that are capable of converting light into spikes monolithically represent a step forward in the artificial visual system with high biological plausibility. More importantly, the large response range corresponding to three wavelengths of light makes it possible to discriminate 'colors'. An ultralow power consumption of ≤400 pW per spike in response to visible lights can be achieved. As a proof-of-concept, such devices were implemented to mimic color perception. The devices with mixed-color selectivity showed a higher recognition accuracy to MNIST hand-written digits with a color-blind test style in comparison with the devices without such selectivity. These results reveal the great potential of such devices for constructing an artificial visual system with high energy efficiency and high biological plausibility. Future improvement could be devoted to the manufacturing of large-scale arrays with the capability to process images or even videos with high resolution with the aid of necessary peripheral circuits. With further integration with spiking neural networks, energy efficiency, and accuracy might be improved. Furthermore, a more biologically plausible system would be available by translating the devices into a flexible/stretchable form, just like the biological cones in the retina.

## Methods

### Fabrication of the vertically integrated VISCP

The VISCP is built based on a vertically integrated configuration of ITO/Ta$_2$O$_5$/Ag/IGZO/ITO. First, an ITO bottom electrode with a thickness of 100 nm was obtained on a pre-cleaned silicon substrate by radio frequency (RF) magnetron sputtering for 10 min in a pure argon ambient at 0.8 Pa using an ITO target (90 wt% In$_2$O$_3$ and 10 wt% SnO$_2$). A Ta$_2$O$_5$ (80 nm) switching layer was deposited by magnetron sputtering for 30 min using a Ta$_2$O$_5$ target (100 wt% Ta$_2$O$_5$). During the sputtering process, the RF power and the Ar:O$_2$ ratio were 100 W and 30:2, respectively. The patterned circular Ag intermediate electrodes (200 µm in diameter) were deposited on the Ta$_2$O$_5$ switching layer by thermal evaporation with a metal mask process. Then, the 55 nm IGZO sensitized layers (120 µm in diameter) were deposited on Ag (100 nm) electrodes by RF magnetron sputtering using IGZO targets (In:Ga:Zn = 2:2:1 atom ratio). During the sputtering process, the RF power and the Ar:O$_2$ ratio were 100 W and 15:15 for 30 min, respectively. Finally, the ITO top electrodes (70 µm in diameter) were deposited by RF magnetron sputtering using an ITO target in a pure argon atmosphere at 0.8 Pa for 10 min.

### Device characterization

The threshold switching characteristics and encoding performance of the devices were tested by the Fs-Pro PX500. For the VISCP measurements, fiber-coupled laser modules (CNI, Laser PGL-FC-360, Laser PGL-FC-405, Laser PGL-FC-532) were used to apply persistent light at 360, 405, and 532 nm on top of the device. The switching times of the devices were performed by using a Keithley 4200 semiconductor parameter analyzer. The capacitance of Ta$_2$O$_5$ was measured by a HIOKI IM 3533-01 LCR instrumentation impedance analyzer. Cross-sectional images of ITO/Ta$_2$O$_5$/Ag/IGZO/ITO optical encoding nerve components were measured by field emission scanning electron microscopy (JEOL, JSM-7000F) for measurement. All the devices in this work were measured in a probe station in the atmospheric environment. The humidity and temperature are ~50% RH and ~300 K.

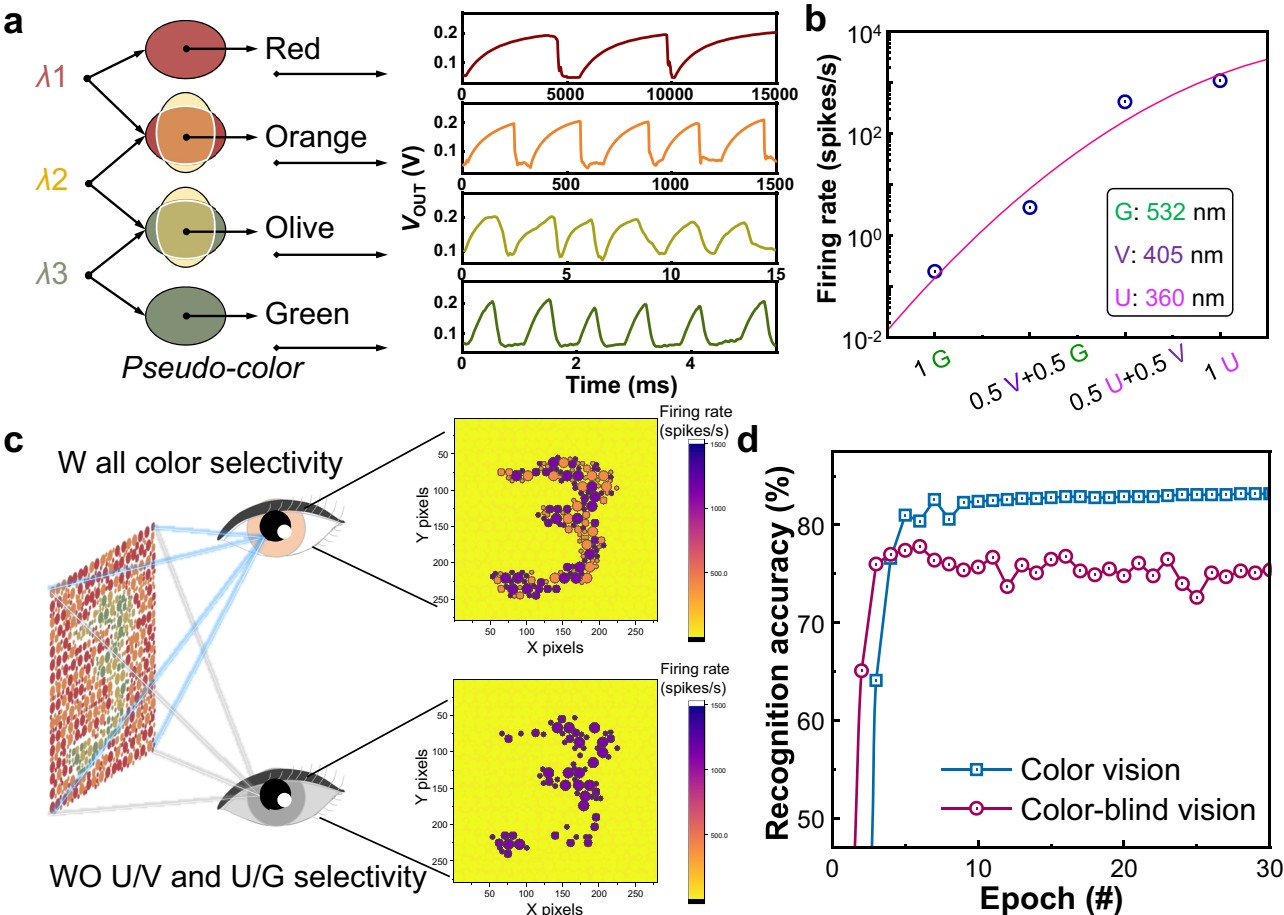

**Fig. 4 | Color-mixed pattern recognition. a** With different combinations of wavelengths (red, orange, olive, and green), the oscillation waveform diagram. **b** The characteristic of firing rate in the case of different ratios of wavelength components. **c** Comparison of color blindness recognition results with and without color selectivity. **d** The evolution of recognition accuracy as a function of training epochs w/wo color selectivity.

## Data availability
All data that support the findings of this study are present in the paper, Supplementary Materials, and associated files. Source data are provided in this paper.

## Code availability
Code from this study (MATLAB scripts) is available from the corresponding author upon request.

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

## Acknowledgements
The authors are grateful for the financial support from the National Key R&D Program of China (2019YFB2205400 (Q.W.) and 2021YFA1202600 (C.W.)), and the National Natural Science Foundation of China (Grant no. 62074075 (Q.W.), no. 61921005 (Q.W.) and no. 62174082 (C.W.)).

## Author contributions
C.W. and Q.W. conceived and supervised the experiments. C.W., Q.W., C.C., and X.W. proposed ideas and formulated overarching research goals. X.W., C.C., L.Z., and K.S. performed the device fabrication and electrical measurements. C.W. and B.P. built the software package. X.W., H.L., and C.F. carried out the characterization of device cross sections. Y.Z., H.M., and S.K. assisted in the fabrication of samples. X.W. and Y.Z. performed the capacitance test. X.W. and C.C. contributed equally to this work. All authors discussed the results and commented on the paper.

## Competing interests
The authors declare no competing interests.
