## [Peer Review File · Nature Communications]

REVIEWER COMMENTS

Reviewer #1 (Remarks to the Author):

This work proposed an oxide-based spiking cone photoreceptor in a vertical integrated structure. Such design is capable of encoding persistent light into a spike train, capturing the very core of a biological photoreceptor. More importantly, the oxygen-rich IGZO is able to selectively respond to different wavelengths of lights, making it possible for mimicking the color-perception of the cone-type photoreceptor. In this case, a high biologically plausible and high density (in potential) spiking cone photoreceptor is available. I believe it would open a new road to toward high energy-efficient and robust visual processing system, and might also nourish the development of dynamic vision sensor and ocular prosthesis. Therefore, I would like to recommend for acceptance after a major revision. Below are my detailed suggestions and questions.

1. An explanation about the light selectivity of the IGZO-based light sensor is necessary, which will make it clear to broader readership.
2. Actually, the IGZO might not be a perfect sensor to visible lights. So, some discussion on how to optimize the light sensing component in terms of materials, structures, and devices are suggested.
3. The recognition accuracy in Fig. 4 is not too high. What's the possible reasons? Whether it's possible for achieving a higher result by optimizing the algorithms?
4. In method part, authors should clarify the measurement conditions like: 1) does this device measured in a vacuum/air or close/open environment? 2) how about the humidity or concentration of shielding gas?

Reviewer #2 (Remarks to the Author):

This manuscript describes the vertically integrated spiking cone photoreceptor (VISCP) in ITO/Ta2O5/Ag/IGZO/ITO device configuration which is capable of converting light with different wavelengths into spike trains. In this device configuration, Ag/IGZO/ITO used as photoresistor and ITO/Ta2O5/Ag as threshold switching memristor for the spike-encoder. The authors have demonstrated the VISCPs with an ultralow power consumption of ≤ 400 pW per spike in visible light. The authors also demonstrated the color-blind test simulations to show image recognition. The manuscript shows certainly interesting results, but there are some major issues as mentioned below. Therefore, this manuscript may be considered for publication after major revision.

1. The authors have integrated the photoresistor with the previously published device on visual depth perception (<https://onlinelibrary.wiley.com/doi/full/10.1002/adma.202201895>). The main concern in this manuscript is that the detection of different wavelengths using an indium-gallium-zinc-oxide (IGZO)-based photodetector with a 3.5 eV bandgap. The change in photoresistances is due to many different reasons, we can't directly correlate it with wavelength change. Explain it.
2. Why this device is called vertical integration? maybe a better term is "monolithic"
3. How did the negative differential resistance (NDR) effect helps in device relaxation? At the same time, the authors also claimed that the memristor device(ITO/Ta2O5/Ag) switches back from low resistive to high resistance through the rapture of conductive filaments.
4. It is very much necessary that the authors should compare their device performance with the reported literature on the same area.
5. How about the device to device variation? How many devices did the authors fabricated and tested? It is necessary to include the device-to-device variation data.
6. What is the exact protocol that constitutes the "training" set in color-blind image recognition? How data for different mixed colors were generated through the spikes, is not clear. Please explain.
7. There are many grammatical and writing errors in the manuscript. Please check out the manuscript throughout and correct them.

Reviewer #3 (Remarks to the Author):

Summary:

In this paper, the authors demonstrate a vertically integrated spiking cone photoreceptor (VISCP) array that can transduce lights into spikes, wherein VISCPs have an ultralow power consumption that is very close to biological cones. In addition, the authors demonstrate the applicability of VISCP's discriminability for different combinations of lights toward recognition tasks including the color-blind test. The work is interesting; however, I have major and minor comments which do not allow me to recommend the publication in its current form.

Major comments:

1) The work spans from the material/device level all the way up to the circuit architecture and algorithm level. However, it is also this broadness that makes judging the work difficult, as the authors combine state-of-the-art technologies to create a more complex system which is difficult to benchmark in its entirety. The authors fail to put the performance of their building-blocks into the context of prior-art, where clearly bench-marking would be possible and desirable.

2) The components that exist in VISCP (spike-encoder and photoreceptor) are materialized independently, in its current form. Considering broad readership of Nature Communications, it is necessary to clarify the role of each element and to make the interaction with each other more specific.

3) There is the absence of any fundamental understanding that ensures the mechanisms on i) the threshold switching of ITO/Ta₂O₅/Ag spike-encoder and ii) the photoresist of Ag/IGZO/ITO photoreceptor. Although the author referred to some mechanism-related papers, it would be more appropriate to perform additional characterization on the mechanisms of the devices. It can improve readers' understanding of the VISCP.

4) In this work, ITO/Ta₂O₅/Ag/IGZO/ITO stack is exploited to implement VISCP; however, insight into the justification for using this stack is not provided. Could only those materials be used to implement VISCP? Or could other materials be suggested as well? The authors should specify the justification for using the materials by linking them to the core operating parameters of VISCP.

5) In its current form, the demonstration and description of the recognition task are vague as follows. Thus, some questions are as follows.

- What is a dependency of light intensity on recognition tasks? In lines 176-177, the authors provide information on the intensity of light (0.5 nW/μm²). However, in real applications, the intensity of light is not fixed. Thus, it would be appropriate to provide some insight into this.

- It seems that there is a lack of specification of the link between parameters of VISCP and recognition tasks. For example, recognition accuracy with respect to intensity and so on.

- The obtained value on recognition accuracy in this work is 75%. This value needs to be improved. In addition, the obtained value on recognition rate should be compared with the state-of-the-art results reported thus far to give justification.

- Using supporting information and/or method section, the authors should provide transparently detailed information and process on recognition tasks with providing specific parameters used in neural networks.

Minor comments:

- 1) Figure 1 in its current form is likely to confuse the reader (e.g., what is the purpose of putting brain images? If there is a purpose, it should be more specific. Not only brain image but other images.). It is recommended to materialize through reorganization or text addition.
- 2) The authors claim that <400 pW per spike is “ultralow power consumption.” Objectivity on this claim should be provided through benchmarking table. Also, how does the ultralow power consumption affect recognition tasks?
- 3) Additional characterization, including Raman, AFM, and so on, on the device should be provided to give the reader an idea on various aspects.
- 4) In line 117-118, the author claims the NDR effect provides the basis for a relaxation spike-encoder. It is required direct description or provision of experimental results on “relaxation” is required, not just providing references.
- 5) How does the scale (thickness and area) of the devices (spike-encoder and photoreceptor) affect performances? What is the minimum feasible scale and what is the corresponding performance of the device?
- 6) The authors claim “array” is implemented in this work, also as state in the title of this article. However, information regarding “array” is not provided. What is the scale? uniformity? association with recognition tasks?

Reviewer #1 (Remarks to the Author):

This work proposed an oxide-based spiking cone photoreceptor in a vertical integrated structure. Such design is capable of encoding persistent light into a spike train, capturing the very core of a biological photoreceptor. More importantly, the oxygen-rich IGZO is able to selectively respond to different wavelengths of lights, making it possible for mimicking the color-perception of the cone-type photoreceptor. In this case, a high biologically plausible and high density (in potential) spiking cone photoreceptor is available. I believe it would open a new road to toward high energy-efficient and robust visual processing system, and might also nourish the development of dynamic vision sensor and ocular prosthesis. Therefore, I would like to recommend for acceptance after a major revision. Below are my detailed suggestions and questions.

1. An explanation about the light selectivity of the IGZO-based light sensor is necessary, which will make it clear to broader readership.

Response: The band gap of perfect IGZO films is about 3.5 eV, which can absorb high-energy photons (e. g., UV light). Such absorption enables the valence electrons to gain enough energy to break the bonding with the parent atom and they jump into the conduction band. This process would induce the so call photocurrent in the semiconductor and an increase in the total conductance of IGZO. So, the electrons can obtain more energy from higher light intensity than from lower light intensity, which induces a larger increase in conductance. As electrons can gain less energy from light with a longer wavelength, the light with longer wavelength would induce a lower increase in the conductance of IGZO.

Most of the electrons in valence cannot gain enough energy to break the bonding with the parent atom with the light of a much longer wavelength (normally >400 nm). However, there are still several mechanisms that can induce photoconductance in this case. One possible mechanism is due to the impurity of the IGZO, some traps and impurities generated during the fabrication process can still absorb energy from the light with low wavelength and induce a photoconductance. Another mechanism might be due to the photothermal effect. The irradiation of the light with a longer wavelength would generate a certain amount of heat on the semiconductor. Some of the valence band electrons can gain enough energy in the form of heat to break the bonding with the parent atom.

The last two mechanisms are much weaker than the first mechanism because the

photoconductance is dependent on the amount of energy that can be absorbed by the semiconductor. Fig. R1 shows the optical absorption spectra of the IGZO film. It clearly shows that the light absorption of the IGZO film decreases with increasing wavelength. Therefore, light with different wavelengths can induce a significant difference in conductance, which is the main reason for the light selectivity of IGZO film. Lights with three wavelengths were used in this work. The wavelength gaps among the three lights are large enough to enable the differentiation by the IGZO-based photosensor as shown in Fig. 2e and the whole artificial cone photoreceptor as shown in Fig. 3c.

This content has been included in the revised manuscript and highlighted in red.

Fig. R1. Absorption spectrum of the IGZO film.

2. Actually, the IGZO might not be a perfect sensor to visible lights. So, some discussion on how to optimize the light sensing component in terms of materials, structures, and devices are suggested.

Response: Yes, we agree that IGZO is not a perfect sensor for visible lights due to its wide bandgap. In principle, semiconductors with a bandgap of 1.6-3.1 eV should be suitable for visible light sensing. There are also a lot of emerging materials like perovskite [Adv. Mater. 2021, 33, 2102300.], metal-organic frameworks (MOF) [Adv. Mater. 2017, 29, 1605071] and conjugated semiconducting polymers [Adv. Mater. Technol. 2021, 6, 2000857.] can have a lower bandgap and visible light sensing property. Materials with specific nano structures such as nanopores, nanowires, and quantum dots are also good candidates for the optimized visible light sensors. The resistance of such photosensors should match the requirement of the circuit as shown

in Fig. 3a, while the semiconductor with a narrow bandgap always with high conductivity (corresponding to high dark current). Therefore, a possible optimization method is to combine IGZO with other narrow bandgap semiconductors or nanomaterials such as silver nanoparticles, graphene dots, and quantum dots to enhance the visible light response [ACS Appl. Mater. Interfaces 7, 19666, 2015] [IEEE Electron Device Lett. 36, 44, 2015] [Nanoscale Research Letters 17, 102 (2022)]. For example, Ref. [Nanoscale Research Letters 17, 102 (2022)] proposed a CdSe quantum dot/IGZO hybrid phototransistor which is sensitive to visible light of 635 nm.

3. The recognition accuracy in Fig. 4 is not too high. What's the possible reasons? Whether it's possible for achieving a higher result by optimizing the algorithms?

Response: Yes, the accuracy in Fig. 4 is not too high, in comparison to many synaptic devices in dealing with the recognition task to MNIST. One main reason is due to the fact that the handwritten digits with color-blind test style (CBTS-MNIST) used in this work are much more complex than standard MNIST (s-MNIST). The image size of CBTS-MNIST is 100 times larger than s-MNIST. What's more, each handwritten digit of the CBTS-MNIST consists of circles with a random combination of four gray levels (corresponding to the spike rate) and four radiuses.

On the other hand, a simple three-layer artificial neural network (ANN) was used, which is not a powerful tool for achieving high recognition accuracy. While the high recognition accuracy is not the main goal of this work. This work is to demonstrate an artificial cone photoreceptor that is able to convert persistent light into spikes, which well mimics the information encoding scheme of its biological counterpart. The recognition results provided in 4d is to show the superiority of the artificial photoreceptor devices with color selectivity in comparison with devices without color selectivity.

To further improve the recognition accuracy, we attempted to use a convolutional neural network (CNN) for CBTS-MNIST recognition, and a higher accuracy of 83.2% can be achieved (Fig. R2). The details of the CNN have been provided in the Supplementary information.

Fig. R2. The recognition accuracies based on CNN algorithm.

4. In method part, authors should clarify the measurement conditions like: 1) does this device measured in a vacuum/air or close/open environment? 2) how about the humidity or concentration of shielding gas?

Response: All the devices in this work were measured in a probe station at the atmospheric environment. The humidity and temperature are ~50% RH and ~300 K.

Reviewer #2 (Remarks to the Author):

This manuscript describes the vertically integrated spiking cone photoreceptor (VISCP) in ITO/Ta₂O₅/Ag/IGZO/ITO device configuration which is capable of converting light with different wavelengths into spike trains. In this device configuration, Ag/IGZO/ITO used as photoresistor and ITO/Ta₂O₅/Ag as threshold switching memristor for the spike-encoder. The authors have demonstrated the VISCPs with an ultralow power consumption of ≤ 400 pW per spike in visible light. The authors also demonstrated the color-blind test simulations to show image recognition. The manuscript shows certainly interesting results, but there are some major issues as mentioned below. Therefore, this manuscript may be considered for publication after major revision.

1. The authors have integrated the photoresistor with the previously published device on visual depth perception

(<https://onlinelibrary.wiley.com/doi/full/10.1002/adma.202201895>);. The main concern in this manuscript is that the detection of different wavelengths using an indium-gallium-zinc-oxide (IGZO)-based photodetector with a 3.5 eV bandgap. The change in photoresistances is due to many different reasons, we can't directly correlate it with wavelength change. Explain it.

Response: The band gap of perfect IGZO films is about 3.5 eV, which can absorb high-energy photons (e. g., UV light). Such absorption enables the valence electrons to gain enough energy to break the bonding with the parent atom and they jump into the conduction band. This process would induce the so call photocurrent in the semiconductor and an increase in the total conductance of IGZO. So, the electrons can obtain more energy from higher light intensity than from lower light intensity, which induces a larger increase in conductance. As electrons can gain less energy from light with a longer wavelength, light with a longer wavelength would induce a lower increase in the conductance of IGZO.

Most of the electrons in valence cannot gain enough energy to break the bonding with the parent atom with the light of a much longer wavelength (normally >400 nm). However, there are still several mechanisms that can induce photoconductance in this case. One possible mechanism is due to the impurity of the IGZO, some traps and impurities formed during the fabrication process can still absorb enough energy from the light with low wavelength to induce a photoconductance. Another mechanism might be due to the photothermal effect. The irradiation of the light with a longer wavelength

would generate a certain amount of heat on the semiconductor. Some of the valence band electrons can gain enough energy in the form of heat to break the bonding with the parent atom. In some previous reports, IGZO-based phototransistors can respond to the light with a wavelength of ~ 550 nm and even longer [ACS Appl. Mater. Interfaces 2018, 10, 8102–8109] [IEEE ELECTRON DEVICE LETTERS 38, 584, 2017], which are consistent with our deductions.

The last two mechanisms are much weaker than the first mechanism because the photoconductance is dependent on the amount of energy that can be absorbed by the semiconductor. Fig. R3 shows the optical absorption spectra of the IGZO film. It clearly shows that the light absorption of the IGZO film decreases with increasing wavelength. Therefore, the light with different wavelength can induce significant difference in conductance, which is the main reason of light selectivity of IGZO. Lights with three wavelengths were used in this work. The wavelength gaps among the three lights are large enough to enable the differentiation by the IGZO-based photosensor as shown in Fig. 2e and the whole artificial cone photoreceptor as shown in Fig. 3c.

This content has been included in the revised manuscript and highlighted in red.

Fig. R3. Absorption spectrum of the IGZO film.

2. Why this device is called vertical integration? Maybe a better term is “monolithic”

Response: Thanks for your suggestion. This device was fabricated vertically as shown in Fig. 1e, which requires more strict fabrication conditions than that of devices with planar structures. This structure is compact and favorable for further scaling down. What’s more, it is of better similarity to its biological counterpart as shown in Fig. 1b.

There were several reports on artificial sensory neurons/receptors with planar structure. Although the device has also exhibited a step forward the spike-encoding capability, we think the vertical integration could also be one of the advantages in comparison with these artificial sensory neurons/receptors.

3. How did the negative differential resistance (NDR) effect helps in device relaxation? At the same time, the authors also claimed that the memristor device (ITO/Ta₂O₅/Ag) switches back from low resistive to high resistance through the rapture of conductive filaments.

Response: The NDR effect means that as the applied current is increased, the voltage decreases instead:

$$R = \frac{dV}{dI} < 0 \quad (1)$$

As shown in Fig. R4, when the memristor voltage reaches the threshold voltage (V_{TH}), the voltage decreases as the applied current increases, resulting in a negative resistance. The NDR effect is attributed to the formation of the Ag filament. This causes a sharp drop in resistance above the V_{TH} . In turn, the spontaneous rapture of the Ag filament causes the memristor to switch to the high resistance state. The formation and breaking of the Ag filament cause a spontaneous switching between two phases in the relaxation spike-encoder: in the first stage the switching element is in the OFF (high resistance) state and the parallel capacitor is charging, and in the second stage the switching element is in the ON (low resistance) state and the parallel capacitor is discharging. The NDR effect, also known as threshold switching, is the basis of the relaxation spike-encoder.

This content has been included in the revised manuscript and highlighted in red.

Fig. R4 The NDR behavior of the threshold switching memristor (TSM) under current-sweeping mode.

4. It is very much necessary that the authors should compare their device performance with the reported literature on the same area.

Response: Thanks for your constructive suggestion. We added the following table R1 in the revised manuscript, which summarized essential performance including reported artificial visual nerve/photoreceptors and our device. Our work features both spike encoding and color perception. It has a low power consumption (≤ 400 pW per spike) in visible light, similar to the photoreceptors of the human eye, which well mimics the information encoding scheme of its biological counterpart.

Table R1 Summary of the reported basic performance of the artificial visual nerve/photoreceptors and our device.

Artificial visual nerve/photoreceptor	Spike encoding	Spike rate	Power consumption per spike	Light response range	Vision-related functions	Ref.
h-BN/Wse ₂ photoresistor & Wse ₂ transistor	×	-	-	405-655 nm	Colored and color-mixed pattern recognition	Nat. Commun. 9: 5106 (2018)
PTCDI-C8/VOPc light-sensitive element & P(VDF-TrFE)/ P(VP-EDMAEMAES) gated P(IID-BT) transistor	×	-	-	550-850 nm	light intensity and frequency transduction	Adv. Mater. 2018, 1803961
ITO/MoOx/Pd/SiO ₂ /Si	×	-	-	365 nm	Image memorization and	Nat. Nanotech. 14, 776 (2019)

					preprocessing	
PEA ₂ MA ₂ Pb ₃ I ₁₀ -based photoresistor & ITO transistor	×	-	-	Solar light	Visual-haptic fusion	Nat. Commun. 11, 4602 (2020)
Ta/InGaZnO ₄ /Pt & Pt/NbO _x /Ta	√	~1.4-5 × 10 ⁶ spike/s	~1 mW @365 nm ~ 2 mW @254 nm	254-365 nm	UV image segmentation	Nano Lett. 2020, 20, 8015–8023
Commercial photoresistor & Ag/TaO _x /ITO	√	1-200 spike/s	0.5 μW @532 nm	532 nm	Visual depth perception	Adv. Mater. 34, 2201895 (2022)
p-i-n perovskite optoelectronic device	×	-	-	435-700 nm	Excitatory and inhibitory light-mediated synaptic functions	Matter 5, 1578-1589 (2022)
MoS ₂ transistor	×	-	-	660 nm	Visual adaptation	Nat. Electron. 5 84 (2022)
Ti/Au/GaO _x /SiO ₂ /Si & TiN/TaO _x /HfO _x /TiN	×	-	-	254 nm	Latent fingerprint identification	Nature Communications 13:6590 (2022)
ITO/IGZO/Ag/TaO _x /ITO	√	0.1-1200 spike/s	210 nW @360 nm 400 pW @405 nm 4.1 pW @532 nm	360-532 nm	Color perception	This work
Photoreceptor in human eye	√	~1-1000 spike/s	250 pW @Dark 50 pW @Bright	400-780 nm	Visual perception	PNAS, 102 (39) 14063-14068 (2005) Clinical and Experimental Ophthalmology 45, 730-741 (2017)

5. How about the device to device variation? How many devices did the authors fabricated and tested? It is necessary to include the device-to-device variation data.

Response: We prepared 1,000 VISCPs and data from 100 of them were randomly selected and used to demonstrate the results. The device-to-device distributions of switching voltages for the TSM part and the whole VISCP are summarized as follows. There is a certain amount of variation (Fig. R5 a) among the TSM in terms of set and reset voltages (corresponding to the amplitude of the output spike train). The Gaussian fits demonstrate the fluctuation of V_{TH} ($\sim 0.27 \text{ V} \pm 0.06$) and V_{HOLD} ($\sim 0.04 \text{ V} \pm 0.017$).

The frequency statistics of the different responses of the VISCP device corresponding to the three wavelengths are shown in Fig. R5 b. The spike frequencies of the VISCPs can be clearly distinguished at the three illumination wavelengths. We have added these device-to-device variation data to the Supplementary information (SI).

We've added this content in the revised SI.

Fig. R5 a, Device-to-device distributions of switching voltages extracted from 100 randomly selected devices. b, Spike frequency statistics at three illumination wavelengths. The light intensity was kept constant at $0.5 \text{ nW}/\mu\text{m}^2$.

6. What is the exact protocol that constitutes the “training” set in color-blind image recognition? How data for different mixed colors were generated through the spikes, is not clear. Please explain.

Response: The process flow was shown in Fig. R6. Each pixel in a handwritten digit with a color-blind test (CBTS-MNIST) represents a light source with a certain combination of wavelengths and an intensity of $0.5 \text{ nW}/\mu\text{m}^2$. Firstly, a CBTS-MNIST image was projected to the device array with one-to-one correspondence. Then, each artificial cone photoreceptor can convert the light with mixed colors into a train of spikes at a certain rate which is simulated based on the data in Fig. 4b. In this case, the image of the CBTS-MNIST digit can be remapped into an image with pixels of spiking rate. Finally, the remapped CBTS-MNISTs were binarized (with a threshold of 423 spike/s). Such binarized CBTS-MNIST images were used as the training and test set. This figure has been included in the revised SI.

Fig. R6 The process flow of transforming CBTS-MNIST handwritten digit to the training/test set.

7. There are many grammatical and writing errors in the manuscript. Please check out the manuscript throughout and correct them.

Response: We have checked the manuscript throughout and amended these errors and typos as many as possible. We've also highlighted these errors in red.

Reviewer #3 (Remarks to the Author):

Summary:

In this paper, the authors demonstrate a vertically integrated spiking cone photoreceptor (VISCP) array that can transduce lights into spikes, wherein VISCPs have an ultralow power consumption that is very close to biological cones. In addition, the authors demonstrate the applicability of VISCP's discriminability for different combinations of lights toward recognition tasks including the color-blind test. The work is interesting; however, I have major and minor comments which do not allow me to recommend the publication in its current form.

Major comments:

1) The work spans from the material/device level all the way up to the circuit architecture and algorithm level. However, it is also this broadness that makes judging the work difficult, as the authors combine state-of-the-art technologies to create a more complex system which is difficult to benchmark in its entirety. The authors fail to put the performance of their building-blocks into the context of prior-art, where clearly bench-marking would be possible and desirable.

Response: Our work is focused on the device level, and the circuit/algorithm were served as proof of concept. We have demonstrated an artificial cone photoreceptor that is able to convert persistent light into spike trains at a certain rate according to the input wavelength, which well mimics the information encoding scheme of its biological counterpart. As a proof-of-concept for cone photoreceptors, such devices were implemented for mimicking color perception. The devices with mixed-color selectivity showed a higher recognition accuracy to MNIST handwritten digits with a color-blind test style in comparison with the devices without such selectivity. We added the following table R2 in the revised manuscript, which summarized essential performance including reported artificial visual nerve/photoreceptors and our device. Our device is the first reported artificial visual nerve/photoreceptor with both spike-encoding and color perception. It has a much lower power consumption than previous works and has a very close power consumption (≤ 400 pW per spike) to the photoreceptors of the human eye in response to visible light.

Table R2 Summary of the reported basic performance of the artificial visual nerve/photoreceptors and our device.

Artificial visual nerve/photoreceptor	Spike encoding	Spike rate	Power consumption	Light response	Vision-related functions	Ref.
----------------	------------	-------------------	----------------	--------------------------	------

			per spike	range		
h-BN/WSe ₂ photoresistor & WSe ₂ transistor	×	-	-	405-655 nm	Colored and color-mixed pattern recognition	Nat. Commun. 9: 5106 (2018)
PTCDI-C8/VOPc light-sensitive element & P(VDF-TrFE)/ P(VP-EDMAEMAES) gated P(IID-BT) transistor	×	-	-	550-850 nm	light intensity and frequency transduction	Adv. Mater. 2018, 1803961
ITO/MoO _x /Pd/SiO ₂ /Si	×	-	-	365 nm	Image memorization and preprocessing	Nat. Nanotech. 14, 776 (2019)
PEA ₂ MA ₂ Pb ₃ I ₁₀ -based photoresistor & ITO transistor	×	-	-	Solar light	Visual-haptic fusion	Nat. Commun. 11, 4602 (2020)
Ta/InGaZnO ₄ /Pt & Pt/NbO _x /Ta	√	~1.4-5 × 10 ⁶ spike/s	~1 mW @365 nm ~ 2 mW @254 nm	254-365 nm	UV image segmentation	Nano Lett. 2020, 20, 8015–8023
Commercial photoresistor & Ag/TaO _x /ITO	√	1-200 spike/s	0.5 μW @532 nm	532 nm	Visual depth perception	Adv. Mater. 34, 2201895 (2022)
p-i-n perovskite optoelectronic device	×	-	-	435-700 nm	Excitatory and inhibitory light-mediated synaptic functions	Matter 5, 1578-1589 (2022)
MoS ₂ transistor	×	-	-	660 nm	Visual adaptation	Nat. Electron. 5 84 (2022)
Ti/Au/GaO _x /SiO ₂ /Si & TiN/TaO _x /HfO _x /TiN	×	-	-	254 nm	Latent fingerprint identification	Nature Communications 13:6590 (2022)
ITO/IGZO/Ag/TaO _x /ITO	√	0.1-1200 spike/s	210 nW @360 nm 400 pW @405 nm 4.1 pW @532 nm	360-532 nm	Color perception	This work
Photoreceptor in human eye	√	~1-1000 spike/s	250 pW @Dark 50 pW @Bright	400-780 nm	Visual perception	PNAS, 102 (39) 14063-14068 (2005) Clinical and Experimental

						Ophthalmology 45, 730-741 (2017)
--	--	--	--	--	--	-------------------------------------

2) The components that exist in VISCP (spike-encoder and photoreceptor) are materialized independently, in its current form. Considering broad readership of Nature Communications, it is necessary to clarify the role of each element and to make the interaction with each other more specific.

Response: Thanks for your suggestion. The VISCP is built with five layers in a vertically integrated configuration, as shown in Fig. 1e. Its equivalent circuit is shown in Fig. 3a. To make a clearer illustration, we combined the two shown in Fig. R7. The top three layers (ITO/IGZO/Ag) are the IGZO photoresistor. The bottom three layers (Ag/Ta₂O₅/ITO) are the Ta₂O₅-based threshold switching memristor (TSM) and can function as a spike encoder. The two devices are connected in series through the middle silver layer. More detailed, the first layer from up to bottom is an indium oxide (ITO) film which is served as the transparent electrode. A voltage bias was applied in this electrode to obtain the spiking output. The input light can pass through the ITO electrode and penetrate into the second layer. The second layer is an indium-gallium-zinc-oxide (IGZO) film in which the conductance can be tuned depending on the intensity and wavelength of the light input (Fig. 2e and 2f). The third layer is a conducting layer of silver which is to connect the two functional devices (IGZO-based photoresistor and Ta₂O₅-based threshold switching memristor). This silver layer is also served as the output terminal where the output spikes can be measured. What's more, the layer can be served as Ag source enabling the formation of the Ag-based conducting path through the fourth layer. The fourth layer is the tantalum oxide (Ta₂O₅) which is served as the medium for silver migrants. The formation/rupture of a silver-based conducting path can occur in the Ta₂O₅ layer under a certain voltage as illustrated in Fig. 2a, inducing a sudden change in conductance of the Ta₂O₅-based threshold switching memristor. The fifth layer is the ITO transparent electrode. This electrode is connected to the ground, which is also served as the substrate for the whole device. When V_{Bias} is applied and the TSM is in a high resistance state (HRS), the parasitic capacitor (C_P) begins to charge. When the C_P voltage exceeds V_{TH} , the TSM switches to a low resistance state (LRS). As a result, the capacitor is discharged through the on-state memristor. The voltage on the C_P then drops below V_{HOLD} and the TSM returns to HRS. Such a charging and discharging process forms a spike train with a certain

frequency.

We've added this content in the revised Supplementary information (SI).

Fig. R7. The layer-by-layer structure of the proposed device and the equivalent circuits for each component.

3) There is the absence of any fundamental understanding that ensures the mechanisms on i) the threshold switching of ITO/Ta₂O₅/Ag spike-encoder and ii) the photoresist of Ag/IGZO/ITO photoreceptor. Although the author referred to some mechanism-related papers, it would be more appropriate to perform additional characterization on the mechanisms of the devices. It can improve readers' understanding of the VISCP.

Response: i) the threshold switching of ITO/Ta₂O₅/Ag originated from the internal Ag dynamics [Nature Communications 7, 11142 (2016)] [Nat. Electron. 1, 137, 2018]. Under a positive electric field, the Ag ions (which are generally readily oxidized to ions in the ambient environment) can pass from the Ag electrode side to the ITO side and induce a cathodic reduction at the ITO side. Then the Ag bridge then can be formed between Ag and ITO with the increase of applied voltage. This formation would induce a sudden decrease in resistance when the Ag-based conducting pathway is formed. As shown in Fig. R8, the device transitions from a high resistance state (HRS) to a low resistance state (LRS) when the applied voltage reaches V_{TH}. The CFs cannot be maintained and will spontaneously rupture if the voltage is lower than V_{HOLD} due to the interfacial energy minimization [Nature Communications 10, 81 (2019)] [Adv. Mater. 2017, 29, 1701752]. The memristor switches to HRS.

Fig. R8. Current-voltage curves of the ITO/Ta₂O₅/Ag memristor.

ii) The band gap of perfect IGZO films is about 3.5 eV, which can absorb high-energy photons (e. g., UV light). Such absorption enables the valence electrons to gain enough energy to break the bonding with the parent atom and they jump into the conduction band. This process would induce the so call photocurrent in the semiconductor and an increase in the total conductance of IGZO. So, more electrons can obtain energy from higher light intensity than from lower light intensity, which induce a larger increase in conductance. As electrons can gain less energy from light with a longer wavelength, the light with a longer wavelength would induce a lower increase in the conductance of IGZO.

Most of the electrons in valence cannot gain enough energy to break the bonding with the parent atom with the light of a much longer wavelength (roughly >400 nm). However, there are still several mechanisms that can induce photoconductance in this case. One possible mechanism is due to the impurity of the IGZO, some traps and impurities generated during the fabrication process can still absorb energy from the light with long wavelength and induce a photoconductance. Another mechanism might be due to the photothermal effect. The irradiation of the light with a longer wavelength would generate a certain amount of heat on the semiconductor. Some of the valence band electrons can gain enough energy in the form of heat to break the bonding with the parent atom.

The last two mechanisms are much weaker than the first mechanism because the photoconductance is dependent on the amount of energy that can be absorbed by the

semiconductor. Fig. R9 shows the optical absorption spectra of the IGZO film. It clearly shows that the light absorption of the IGZO film decreases with increasing wavelength. Therefore, light with different wavelengths can induce a significant difference in conductance, which is the main reason for the light selectivity of IGZO film. Lights with three wavelengths (i. e., 360, 405, and 532 nm) were used in this work. The wavelength gaps among the three lights are large enough to enable the differentiation by the IGZO-based photosensor as shown in Fig. 2e and the whole artificial cone photoreceptor as shown in Fig. 3c.

Fig. R9. Absorption spectrum of the IGZO film.

4) In this work, ITO/Ta₂O₅/Ag/IGZO/ITO stack is exploited to implement VISCP; however, insight into the justification for using this stack is not provided. Could only those materials be used to implement VISCP? Or could other materials be suggested as well? The authors should specify the justification for using the materials by linking them to the core operating parameters of VISCP.

Response: The materials of construction for spike-encoders and photoresistors are not unique. Besides Ag-based ITO/Ta₂O₅/Ag devices, Mott insulators such as VO₂ [Adv. Mater. Interfaces 2022, 9, 2200394.], NbO_x [Nature 548, 318–321 (2017).] are also used in spike encoders. The threshold switching effect of Mott insulators is caused by the insulator-metal transition (IMT). However, the high resistance state of Mott insulators is mostly below 10⁶ Ω, three orders of magnitude lower than Ta₂O₅. This can

result in high leakage current and power consumption.

For photoresistors, other materials have been reported, such as WSe₂, MoO_x and perovskite. However, they always exhibit high conductivity (corresponding to high dark currents). IGZO photoresistors exhibit high color selectivity, which exhibit significant photoresistance difference with more than one order of magnitude in response to mentioned three lights.

Furthermore, matching the spike encoder resistance range to the photoresistor resistance range is the most important aspect of achieving VISCP. For the output spike trains, the TS device has to switch between high (R_{OFF}) and low (R_{ON}) resistance states. In the charging process, R_{OFF} and R_I should satisfy the following relationship:

$$\frac{R_{OFF}}{R_{OFF} + R_I} \times V_{IN} > V_{TH}$$

To ensure that the TSM switches to the low resistance state, the voltage divided by the TSM should be higher than the turn-on voltage (V_{TH}). In the discharge process, R_{ON} and R_I should comply with the following relationship:

$$\frac{R_{ON}}{R_{ON} + R_I} \times V_{IN} < V_{HOLD}$$

To ensure that the TSM switches to the high resistance state, the voltage divided by the TSM is less than the off voltage (V_{HOLD}).

The VISCP composed of Ta₂O₅ and IGZO achieves both color perception and spike encoding functions. The power consumption in response to visible light is ≤ 400 pW, and the spiking rate is ~ 0.1 -1200 Hz, which is very close to the response range of biological cones. The VISCP has advanced bio-similarity in spiking color perception and energy consumption.

5) In its current form, the demonstration and description of the recognition task are vague as follows. Thus, some questions are as follows.

- What is a dependency of light intensity on recognition tasks? In lines 176-177, the authors provide information on the intensity of light ($0.5 \text{ nW}/\mu\text{m}^2$). However, in real applications, the intensity of light is not fixed. Thus, it would be appropriate to provide some insight into this.

Response: We agree that the light intensity in the real world is not fixed, and the bad quality of images might lead to low recognition accuracy. What's more a sophisticated optical recording/processing system. For example, the camera can obtain good image quality the same as photos taken based on the lab condition, by using both software and

hardware approaches. For example, advanced algorithms for image restoration/calibration and filters/circuits for light compensation/correction, and so on. This work is not focused on the optimization of image quality, so we used fixed light intensity to generate high-quality images. The main goal of this work is to demonstrate a fully-integrated cone-type photoreceptor that is capable of converting persistent light into spikes, resembling the information encoding scheme of its biological counterpart.

- It seems that there is a lack of specification of the link between parameters of VISCP and recognition tasks. For example, recognition accuracy with respect to intensity and so on.

Response: In principle, the recognition accuracy is dependent on the quality of the input image. In our case, the image quality is mainly dependent on the contrast between handwritten digits with background, because the spike rate in response to each pixel was converted to a numerical value before the training process. We plotted the normalized (by using the response to light of 360 nm as the reference) spike rates corresponding to each wavelength in Fig. R10. There is a clear difference between the three wavelengths, which provides high contrast for the image. What's more, contrasts among the three wavelengths remain unchanged when the light intensity is above 0.3 $\text{nW}/\mu\text{m}^2$. This means that the effect of light intensity on recognition accuracy can be neglected within the light intensity range of 0.3-0.5 $\text{nW}/\mu\text{m}^2$.

Fig. R10. The contrast plotted as a function of light intensity. The value of contrast here is the normalized spike rate with the maximum spike rate corresponding to each wavelength.

- The obtained value on recognition accuracy in this work is 75%. This value needs to

be improved. In addition, the obtained value on recognition rate should be compared with the state-of-the-art results reported thus far to give justification.

Response: The recognition accuracy of 75% seems not as high as some state-of-the-art results with respect to the standard MNIST (s-MNIST) dataset. However, the color-blind test style MNIST (CBTS-MNIST) generated in this work is much more complex than standard MNIST, which may result in low recognition accuracy. The image size of CBTS-MNIST is 100 times larger than s-MNIST. What's more, each handwritten digit of the CBTS-MNIST consists of circles with a random combination of four gray levels (corresponding to the spike rate) and four radiuses.

On the other hand, a simple three-layer artificial neural network (ANN) was used, which is not a powerful tool for achieving high recognition accuracy. While the high recognition accuracy is not the main goal of this work. This work is to demonstrate an artificial cone photoreceptor that is able to convert persistent light into spikes, which well mimics the information encoding scheme of its biological counterpart. The recognition results provided in Fig. 4d is to show the superiority of the artificial photoreceptor devices with color selectivity in comparison with devices without color selectivity.

To further improve the recognition accuracy, we attempted to use a convolutional neural network (CNN) for CBTS-MNIST recognition, and a higher accuracy of 83.2% can be achieved (Fig. R11). The details of the CNN have been provided in the Supplementary information and are as follows.

Fig. R11. The recognition accuracies based on CNN algorithm.

A five-layer convolutional neural network (CNN) was used (Fig. R12). The first layer

consists of an input image with dimensions of 280×280 . The input data is then convolved with 16 filters of 7×7 in size and padding of 2, resulting in dimensions of $278 \times 278 \times 16$. The second layer is a Pooling operation with a filter size of 3×3 . Hence the resulting image dimension is $93 \times 93 \times 16$. Similarly, the third layer also involves in a convolution operation with 32 filters of 7×7 in size and padding of 2 followed by a fourth pooling layer with the same filter size of 3×3 . The resulting image dimension is then reduced to $31 \times 31 \times 32$ in size. The fifth layer is a fully connected layer with 10 nodes to identify the label of each treated digit. In this layer, each of the 10 nodes will be connected to the 30752 ($31 \times 31 \times 32$) nodes from the previous layers. The recognition consists of two phases: training and inferencing. The training phase includes two sub-procedures: forward-pass and weight update. The weight of each node is updated based on the back-propagation algorithm. During the inferencing phase, the network was fed up with the test images to identify the label of them, and the recognition accuracy is thus obtained.

Fig. R12. Structure of the five-layer CNN for CBTS-MNIST recognition.

- Using supporting information and/or method section, the authors should provide transparently detailed information and process on recognition tasks with providing specific parameters used in neural networks.

Response: We had provided some details of recognition task in Supplementary Note 3 and Note 4. We've added more details and schemes in the revised manuscript and Supplementary information based on your suggestion.

Minor comments:

1) Figure 1 in tis current form is likely to confuse the reader (e.g., what is the purpose of putting brain images? If there is a purpose, it should be more specific. Not only brain image but other images.). It is recommended to materialize through reorganization or

text addition.

Response: The brain shown in this figure is to illustrate the color vision processing as shown in Fig. R13. Not strictly, the color perception is formed based on the three processing: perceiving, encoding, and processing. An object is perceived by the photoreceptors (cone cell and rod cell) in our eyes. These photoreceptors can convert optical information into spikes. These spikes convey information about the object (e.g., color and brightness, etc.) and can be processed by the neural network in the visual cortex (the red region in the brain, Fig. R13).

Inspired by such information flow, we design an artificial cone-type photoreceptor for perceiving the colorful light and converting it to spikes with a certain rate that is dependent on the light wavelength. Finally, we simulated the recognition of mixed-color images, which is similar to the function of the neural network in the visual cortex.

To make it clearer, we've added annotations for Fig. 1a as shown in Fig. R13 and the revised manuscript.

Fig. R13. The scheme illustrating the information flow of color vision.

2) The authors claim that <400 pW per spike is “ultralow power consumption.” Objectivity on this claim should be provided through benchmarking table. Also, how does the ultralow power consumption affect recognition tasks?

Response: We've summarized the essential performance of reported artificial visual nerves/photoreceptors and our device in Table 1 of the revised manuscript. The power issues are shown in Table R3. As shown in this table, our device shows a power consumption of no more than 400 pW in response to almost all visible light ($\lambda \geq 405$ nm), which is comparable to human eyes. It has the lowest power consumption (200 nW@360 nm) in response to ultraviolet (UV).

The power consumption of the device doesn't affect the recognition accuracy in principle. The declaration of low power consumption is to demonstrate the energy efficiency of our devices, which is one of the main goals of neuromorphic electronics.

As demonstrated in the discussion section, the encoding scheme of information and the separation of sense, memory, and processing in digital systems would lead to a big computational burden as well as high energy consumption. One advantage of our device is the capability that encodes information into spikes as the biological system, which would alleviate the computation burden and reduce the energy consumption in turn. The low power consumption of the device would further reduce the total energy consumption of a neuromorphic system based on the proposed devices.

Table R3 Summary of the reported power consumption of the artificial visual nerve/photoreceptors and our device.

Artificial visual nerve/photoreceptor	Power consumption per spike	Ref.
Ta/InGaZnO ₄ /Pt & Pt/NbO _x /Ta	~1 mW @365 nm ~2 mW @254 nm	Nano Lett. 2020, 20, 8015–8023
Commercial photoresistor & Ag/TaO _x /ITO	500 nW @532 nm	Adv. Mater. 34, 2201895 (2022)
ITO/IGZO/Ag/TaO _x /ITO	210 nW @360 nm 400 pW @405 nm 4.1 pW @532 nm	This work
Photoreceptor in human eye	250 pW @Dark 50 pW @Bright	PNAS, 102 (39) 14063-14068 (2005). Clinical and Experimental Ophthalmology 45, 730-741 (2017).

3) Additional characterization, including Raman, AFM, and so on, on the device should be provided to give the reader an idea on various aspects.

Response: Amorphous IGZO and Ta₂O₅ films were prepared on silicon and X-ray diffraction (XRD) were measured. The XRD patterns are shown in Fig. R14, and only the peak of the Si substrate (69.3) is observed, confirming the amorphous structure of the films. Raman characterization of the films also showed only the peak of Si (520.7), as shown in Fig. R15. The chemical composition of the IGZO and Ta₂O₅ layers was analyzed by X-ray photoelectron spectroscopy (XPS), as shown in Fig. R16. The In 3d, In 3p, Ga 2p, Zn 2p, and O 1s spectra were clearly observed in the IGZO film. The Ta 4f, Ta 4d, Ta 4p, and O 1s spectra were clearly observed in the Ta₂O₅ film. The surface morphology of IGZO and Ta₂O₅ films was investigated by atomic force microscopy (AFM). The smooth surfaces of IGZO and Ta₂O₅ can be clearly observed from the AFM images. As shown in Fig. R17, the roughness of IGZO and Ta₂O₅ is 0.4 nm and 0.2 nm respectively. The smooth surface morphology provides the prospect of providing good vertical stacking interfaces and high-performance VISCP.

The relevant characterization has been added to the revised SI.

Fig. R14. X-ray diffraction (XRD) pattern of sputtered IGZO and Ta₂O₅ films.

Fig. R15. Raman spectra of IGZO and Ta₂O₅ films.

Fig. R16. X-ray photoelectron spectroscopy (XPS) scanning spectra of IGZO and Ta₂O₅ films.

Fig. R17. AFM diagram of IGZO and Ta₂O₅ films.

4) In line 117-118, the author claims the NDR effect provides the basis for a relaxation spike-encoder. It is required direct description or provision of experimental results on “relaxation” is required, not just providing references.

Response: The NDR effect means that as the applied current is increased, the voltage decreases instead:

$$R = \frac{dV}{dI} < 0 \quad (2)$$

The spontaneous switching between two phases in the relaxation spike-encoder: in the first stage the switching element is in the OFF (high resistance) state and the parallel capacitor is charging, and in the second stage the switching element is in the ON (low resistance) state and the parallel capacitor is discharging. As shown in Fig. R18, when the memristor voltage reaches the threshold voltage (V_{TH}), the voltage decreases as the applied current increases, resulting in a negative resistance. The NDR effect shows a sharp drop in resistance above V_{TH} . This causes the relaxation spike-encoder to switch from the first to the second stage.

Fig. R18 The NDR behavior of the threshold switching memristor (TSM) under current-sweeping mode.

5) How does the scale (thickness and area) of the devices (spike-encoder and photoreceptor) affect performances? What is the minimum feasible scale and what is the corresponding performance of the device?

Response: The spiking frequency of the VISCP is determined by the TSM switching voltage, the circuit resistance, and the parasitic capacitance:

$$f = \frac{1}{T_{rise} + T_{fall}} \quad (3)$$

$$T_{rise} = \frac{(V_{TH} - V_{HOLD}) \times C_P \times (R_{OFF} + R_I)}{V_{IN}} \quad (4)$$

$$T_{fall} = \frac{(V_{TH} - V_{HOLD}) \times C_P \times (R_{ON} + R_I)}{V_{IN}} \quad (5)$$

The spike frequency is obtained from equations (3) - (5) as:

$$f = \frac{V_{IN}}{(V_{TH} - V_{HOLD}) \times C_P \times (R_{OFF} + R_{ON} + 2R_I)} \quad (6)$$

The R_{ON} is a much smaller value than the R_{OFF} and the $2R_I$. The frequency can be estimated as follows:

$$f \approx \frac{V_{IN}}{(V_{TH} - V_{HOLD}) \times C_P \times (R_{OFF} + 2R_I)} \quad (7)$$

The resistance and capacitance as a function of area (S) and thickness (D) are:

$$R = \frac{\rho D_1}{S_1} \quad (8)$$

$$C_P = \frac{\epsilon S_2}{4\pi k D_2} \quad (9)$$

The capacitance is proportional to the area and the resistance is inversely proportional. This means that the performance of the VISCP is not affected when the area is reduced in equal proportions. Theoretically, it is possible to reduce the individual VISCPs to the nanometer size in a similar proportion. However, considering the fabrication process, a large reduction in the area may cause the collapse of the vertical structure. In addition, quantum tunneling effects at the nanoscale cause reduced functionality and increased power consumption of the device. In addition, the reduced performance and increased power consumption caused by the quantum tunneling effect make it difficult to fabricate devices at the nanoscale.

Similarly, as the thickness changes, the effects of resistance and capacitance are

balanced out. However, as device thickness increases, higher turn-on voltage is required to form Ag filaments. The current-voltage curves of the spike-encoder at different sputtering times are shown in Fig. R19a. The V_{TH} and V_{HOLD} of the TSM increase with increasing Ta_2O_5 thickness. The difference between V_{TH} and V_{HOLD} also increases with thickness (Fig. R19b). According to equation (7), the spike rate of the VISCP decreases with increasing thickness. Figure R19c shows that the Ta_2O_5 ON/OFF ratio is too weak to meet the circuit requirements when the growth time is less than 15 min. Fig. R20 shows the photoresistor resistance for different IGZO sputtering times in darkness, 532 nm, 405 nm and, 360 nm illumination. Photoresistor resistance increases with IGZO sputtering time. This is in accordance with the basic law of resistance variation in Equation 8.

In conclusion, the variation of the VISCP area does not affect device performance. The spike firing rate of VISCP decreases with increasing thickness when the device remains undamaged.

Fig. R19 a, Current-voltage curves of the Ta_2O_5 -based threshold switching memristor (TSM) of different sputtering times. b, The switching voltage distribution of the TSM with the different sputtering times. c, The current-voltage curve of the TSM at a sputtering time of 10 min.

Fig. R20 The resistance of IGZO devices 1-3 (D1-D3) with different sputtering times. The light intensity was kept constant at $0.5 \text{ nW}/\mu\text{m}^2$.

6) The authors claim “array” is implemented in this work, also as state in the title of this article. However, information regarding “array” is not provided. What is the scale? uniformity? association with recognition tasks?

Response: The VISCP array is prepared on the two-inch silicon wafer as shown in Fig. 1c. The whole array contains 5×5 sub-arrays and each sub-array consists of 8×8 VISCP devices (i. e., 1600 device in total). We’ve tested the 100 devices in the array and the statistical results were shown in Fig. R5. As a proof-of-concept, the pixels of a colored image are directly mapped onto the array of devices one by one (as shown in Fig. R6). While, the corresponding peripheral circuit was not developed as the recognition tasks were implemented by simulations using the parameters extracted from the statistical results. The uniformity might affect the recognition accuracy, while we haven’t verified this. Our main point is to show such oxide-based device can resemble some behaviors of cone-type photoreceptor and it’s of great potential for building a device array similar as the retina.

REVIEWER COMMENTS

Reviewer #1 (Remarks to the Author):

The authors have successfully addressed all my comments in previous round. I don't have further comments.

Reviewer #2 (Remarks to the Author):

The authors have provided satisfactory answers to all questions and have made required corrections/additions to the manuscript except one, the explanation of detection of different wavelengths using an indium-gallium-zinc-oxide (IGZO)-based photodetector with a 3.5 eV bandgap was not convincing at all.

Moreover, the author explained in response to the reviewer question-1 "so, the electrons can obtain more energy from higher light intensity than from lower light intensity, which induces a larger increase in conductance" which is against the fundamental principle of photophysics. The increase in light intensity means the no. of photons increases, if it is absorbed completely, then the no. of photoelectron generation will increase, not electron energy.

Now come to another fundamental issue, it is not reliable to use the device to detect light intensity or wavelength lower than the bandgap of the active material of the device. In this case, the active material bandgap is 3.5 eV, and the authors have shown photoresistance change within 350 nm-600 nm, which is beyond the band edge. If this is true, then any semiconductor materials can be used for wavelength detection. The observed photoresponse in this device was basically due to defects, these defects are going to vary from device to device, which is very difficult to control, so it is not appropriate to correlate photoresistance change with wavelength detection especially lower energy of the band edge of the active materials. Upto the band edge of the active materials, if you have responsivity data with different wavelengths, it is possible to detect light wavelength properly. Please address this issue.

Reviewer #3 (Remarks to the Author):

The careful update of your manuscript is greatly appreciated. I believe that the work is a valuable contribution to the (still) emerging field of the artificial photoreceptor. I believe the new version is suitable for publication.

Reviewer #2 (Remarks to the Author):

The authors have provided satisfactory answers to all questions and have made required corrections/additions to the manuscript except one, the explanation of detection of different wavelengths using an indium-gallium-zinc-oxide (IGZO)-based photodetector with a 3.5 eV bandgap was not convincing at all.

Moreover, the author explained in response to the reviewer question-1 “so, the electrons can obtain more energy from higher light intensity than from lower light intensity, which induces a larger increase in conductance” which is against the fundamental principle of photophysics. The increase in light intensity means the no. of photons increases, if it is absorbed completely, then the no. of photoelectron generation will increase, not electron energy.

Now come to another fundamental issue, it is not reliable to use the device to detect light intensity or wavelength lower than the bandgap of the active material of the device. In this case, the active material bandgap is 3.5 eV, and the authors have shown photoresistance change within 350 nm-600 nm, which is beyond the band edge. If this is true, then any semiconductor materials can be used for wavelength detection. The observed photoresponse in this device was basically due to defects, these defects are going to vary from device to device, which is very difficult to control, so it is not appropriate to correlate photoresistance change with wavelength detection especially lower energy of the band edge of the active materials. Upto the band edge of the active materials, if you have responsivity data with different wavelengths, it is possible to detect light wavelength properly. Please address this issue.

Response: Firstly, sorry for the less rigorous answer made previously. Yes, the increase of light intensity would not lead to the increased energy absorption by electron, but the increased number of photoelectrons.

Secondly, we agree the photoelectric response of the amorphous IGZO films to the light with longer wavelength than ~400 nm should mainly be due to the absorption by defects related to the oxygen vacancies. In amorphous IGZO thin films, defects caused by oxygen vacancies can be controlled by adjusting the oxygen partial pressure during film deposition. Indeed, IGZO photoresponse for the longer wavelength (400-700 nm) has been reported previously [ACS Appl. Mater. Interfaces 2018, 10, 8102–8109]

[IEEE ELECTRON DEVICE LETTERS 38, 584, 2017] [Nano Energy 2019, 62, 772.] [Appl. Phys. Lett. 98, 232102 (2011)] [Appl. Phys. Lett. 99, 093507 (2011)]. Optical analyses and first-principle calculations have shown that the oxygen vacancies lead to a defect energy level of ~ 2.3 eV [Phys. Stat. Sol. (c) 5, 3098 (2008)]. We've shown the absorption spectrum of the IGZO film in last response, and we also provided here as shown in Fig. R1. The data shown in Fig. R1 was obtained from 20 samples of IGZO films deposited under the same conditions. The absorption corresponding to both 405 nm and 532 nm are nonzero values, and there existed clear differences between them.

Fig. R1. Absorption spectrum of the 20 sample of IGZO films.

To further confirm the selectivity to the lights of three wavelengths, the IV curves in the lights with same intensity of $0.5 \text{ nW}/\mu\text{m}^2$ and dark are shown in Fig. R2. There is more than an order of magnitude difference among the three kinds of lights with respect to response currents of the IGZO photoresistor. The IV curves of the IGZO photoresistor with varied light intensity are shown in Fig. R3, demonstrating the increased current with increasing light intensity.

Fig. R2. Current-voltage curves of the IGZO photoresistor in light illumination at 360, 405, 532 nm and in the dark, respectively. The light intensity was kept constant at $0.5 \text{ nW}/\mu\text{m}^2$.

Fig. R3. The photocurrent of the IGZO device varied with the light intensities with wavelengths of 360, 405, and 532 nm, respectively. The light intensities are 0.2, 0.3, 0.4, and 0.5 $\text{nW}/\mu\text{m}^2$.

Lastly, defect engineering is one of the core strategies in controlling oxide semiconductor properties [Mater. Horiz. 2020, 7, 2832] [Vacuum 86, 1313 (2012)] [J. Appl. Phys. 106, 071101 (2009)]. To evaluate the device-to-device variations of the IGZO-based photoresistors, the absorption spectrum and photoresistance of 20 samples were prepared under the same deposition conditions. As shown in Fig. R4, the difference in absorption and photoresistance between lights with wavelengths of 360, 405, and 532 nm, respectively, are much larger than the device-to-device variation corresponding to each light. So, the device-to-device variation raised from the defect absorption is acceptable for experiment related to ‘color selectivity’ in this work. These data have been included in the revised SI.

Fig. R4. a, Device-to-device variations of light absorption with wavelengths of 360, 405, and 532 nm, respectively. **b,** Device-to-device variations of photoresistance with wavelengths of 360, 405, and 532 nm, respectively. The light intensity was kept constant at 0.5 $\text{nW}/\mu\text{m}^2$.

REVIEWERS' COMMENTS

Reviewer #2 (Remarks to the Author):

The authors have carefully addressed all of the concerns and have made required corrections in the manuscript, so the revised manuscript can be accepted for publication at its current form.